# Effects of Target Variables on Interpersonal Distance Perception for Young Taiwanese during the COVID-19 Pandemic

**DOI:** 10.3390/healthcare11121711

**Published:** 2023-06-11

**Authors:** Yi-Lang Chen, Andi Rahman

**Affiliations:** 1Department of Industrial Engineering and Management, Ming Chi University of Technology, New Taipei 24301, Taiwan; m09218051@mail2.mcut.edu.tw; 2Department of Industrial Engineering, Andalas University, Padang 25175, Indonesia

**Keywords:** interpersonal distance, face covering, vaccination, target sex, participant sex

## Abstract

The COVID-19 pandemic has affected not only public health but also people’s daily lives. Among various strategies to prevent infection, mask wearing and vaccination are considered to be the most effective methods; however, they may affect the comfortable interpersonal distance (IPD) for social interactions. In 2023, although the COVID-19 epidemic is considered to be similar to influenza, the public health sector of Taiwan still plans to give each person at least one dose per year, and even two does for special cases such as the elderly; and more than 90% of Taiwanese are still accustomed to wearing masks in public areas. Compared with mask wearing, studies examining the effects of vaccination on IPD are lacking. Therefore, an online survey was conducted in this study to collect the IPD data of 50 male and 50 female participants to elucidate the effects of mask wearing, vaccination, and target sex variables on IPD. The results showed that all variables significantly affected IPD (all *p* < 0.001). The effect of masks on IPD (49.1 cm) was slightly greater than that of vaccination (43.5 cm). The IPDs reported for wearing and not wearing masks were 145.7 and 194.8 cm, respectively, and those for vaccinated and unvaccinated were 148.5 and 192.0 cm, respectively. Regardless of participant sex, the IPDs for the female targets were significantly shorter than those for the male targets, which was consistent with the results of previous studies. Although mask wearing and vaccination are functionally different in nature, the findings indicate that the effects of both on IPD are nearly identical, jointly shortening IPD to approximately 93 cm. This implies that not only masks but also vaccination could lead to the shortening of IPD and may cause challenges in the prevention and control of COVID-19 transmission.

## 1. Introduction

The COVID-19 pandemic and government guidelines aimed at curbing transmission have changed our daily lives. The term “social distancing” has become familiar and entered daily speech [1,2]. Many governments have imposed policies promoting physical distancing to reduce virus transmission. The World Health Organization recommends numerous methods that have been demonstrated to prevent the transmission of COVID-19, including frequent hand washing, social distancing, and mask wearing, particularly in public spaces [3]. Maintenance of a 1.5 m distance from others in public areas has been enforced because respiratory viruses such as coronaviruses and influenza infect people through droplet inhalation [4].

Chu et al. [5] reported that maintaining a distance of over 1 m reduces the risk of infection. Surgical masks have also been recommended for self-protection [6]. Wearing a face mask has been recognized as an effective method to limit the transmission of COVID-19 [5,7,8]. Despite their effectiveness at preventing transmission, social distancing and isolation policies may affect psychological health [9,10]. People’s facial expressions are hidden when wearing a face mask, which in turn affects their feelings and cognitions during social interactions [11]. People globally have become accustomed to wearing face masks and maintaining an appropriate social distance in daily life. The short-term effects of these measures include discomfort, heightened arousal, and limited social signaling [12]. However, the long-term effects of violating preferred norms of interpersonal distance (IPD) remain undetermined.

The concept of IPD was first proposed by Hall [13]. Interpersonal distance refers to the minimum comfortable distance that people maintain between themselves and others, and it has been widely investigated in psychology, environmental design, and other behavioral sciences. Factors affecting IPD include personal characteristics, target features, and environmental conditions [3]. Studies have indicated that women maintain a greater IPD than do men [3,14,15,16]. Lee and Chen [3] also reported that an interaction effect between region and mask wearing influenced IPD perception, and that Taiwanese participants maintained a significantly greater IPD than did Mainland Chinese participants. IPD perceptions also differ by ethnicity [17,18]. The influence of target features such as sex [19] and facial expression [20] on IPD has also been investigated. These findings have elucidated the effects of psychology, social interaction, and environmental factors on IPD. In general, people maintain a larger IPD to protect themselves when they feel threatened or insecure. Notably, before the COVID-19 pandemic, people wearing face masks were perceived to be risky and dangerous [21], and this gave rise to psychological barriers [22]. Masked people may be labeled as unhealthy, and thus give people the subconscious impression that they should not be approached. Conversely, during the COVID-19 pandemic, people lowered their vigilance during interpersonal contact because of perceived safety, and thus shortened their IPD [1,3,11,23].

Vaccination is another method of COVID-19 prevention; however, vaccination is less widely supported than social distancing and mask wearing [24,25,26,27]. From a public health perspective, vaccination can indeed reduce transmission and mortality rates [28,29,30]. Because COVID-19 vaccine hesitancy primarily arises from adverse events following immunization [31,32,33], people still generally have positive views of vaccination, particularly when case numbers are still rising [34], and when quarantine measures are in place [35]. For example, in Taiwan, by the end of August 2022, the percentages of the population who had received their first, second, and third doses of COVID-19 vaccines were 93%, 87%, and 72%, respectively (http://covid-19.nchc.org.tw, accessed on 12 October 2022). Vaccination does not guarantee total protection against COVID-19; this is particularly true for new variants. Therefore, health officials have continuously implored the public to take extra precautions against the virus and continue to wear masks, wash hands, and observe physical distancing even after vaccination [31]. This complicates IPD-related research that only focuses on effect of mask wearing. In the first quarter of 2023 in Taiwan, although the COVID-19 epidemic is considered to be similar to influenza, the government still plans to give each person at least one dose per year, and in special cases such as the elderly, two doses are required. Recently, in May 2023, Taiwan experienced the fourth peak of the epidemic, and mask policy was thus temporarily adjusted. At present, more than 90% of Taiwanese are still accustomed to wearing masks in public areas.

The pandemic has led to a proliferation of studies exploring IPD in various contexts. Some of these have focused on the role of mask wearing [11] and target factors [3] in the perception of others, and some others have concentrated on the explicit temporal components of IPD changes before and after the pandemic [12]. Although factors influencing IPD have been thoroughly investigated (e.g., mask wearing, target sex, and participant sex), to the best of our knowledge, no study has investigated the effect of vaccination on IPD. Unlike other preventive measures, the main purpose of vaccination is to prevent recipients from becoming infected and to reduce mortality risk after diagnosis [28,29,30]. We hypothesized that people would irrationally reduce their perceived risk level when facing vaccinated individuals, thereby reducing IPD. This study thus examined the effects of vaccination, mask wearing, and target sex variables on IPD perception for both male and female groups. Understanding the changes in IPD in relation to these variables, particularly vaccination, and their interactions may provide useful insights into human social behavior during the COVID-19 pandemic.

## 2. Methods

### 2.1. Participants

A total of 100 young male and female participants (50 each) were recruited from a university in New Taipei (Taiwan) for an online survey. They were all university students, and the main reason for focusing on the young population in the study was that they are relatively socially active compared to other age groups. All participants received complete vaccinations (three or more doses, as required), and they reported no cognitive or psychological problems. Exclusion criteria for participation in the study included: insufficient vaccinations, cold symptoms, or difficulty in operating the mouse. In the test, the mean (standard deviation) age for men (N = 50) and women (N = 50) were 20.6 (1.8) and 20.9 (1.9) years, respectively. The data were collected from February to March 2022. During this period, it happened that Taiwan was still in the severe stage of the COVID-19 epidemic. All participants were right-handed, and were not familiar with the targets in the experiment. Participants were fully informed of the testing procedure and were asked to sign a consent form before data collection. The study was approved by the Ethics Committee of Chang Gung University, Taiwan.

### 2.2. Experimental Design

We conducted a cross-sectional study to examine IPD under various testing conditions. Because of the COVID-19 pandemic, an online survey was used to collect data on IPD to avoid human-to-human transmission [3,19,36]. The online survey was adapted from the paper-and-pencil test used in the studies of Hayduk [37] and Xiong et al. [38]. An online survey is an effective tool for collecting data on IPD, and is widely used in clinical and practical investigations [14]. During the test, each participant was asked to determine the IPD under eight test combinations that comprised of target-sex (man or woman), mask (with and without mask), and vaccination (vaccinated or unvaccinated) variables. Thus, 2400 data samples (100 participants × 2 target sexes × 2 mask conditions × 2 vaccination statuses × 3 repetitions) were collected. Among the three repetitions, the closest two IPD measures were averaged for analysis. The experimental design was counterbalanced across the sexes, and the testing trials were presented one by one for IPD judgment and were randomly arranged.

### 2.3. Experimental Setting

During the test, a computer with the Axure RP rapid prototyping tool (https://www.axure.com/a/rapid-prototyping-tool, accessed on 12 October 2022, Axure Software Solutions, San Diego, CA, USA) was used to conduct the survey. Participants were instructed to use the cursor to move the virtual subject (avatar) toward the target (Figure 1). When a participant started to move the avatar, the arrow guiding the movement direction between the two avatars was hidden to avoid influencing the participant’s distance judgment. That is, no cue was provided for the distance between the two avatars during IPD determination, except for the changes in spatial perception caused by moving the avatar. The participants were asked to imagine and then determine the IPD by moving the avatar to a position that still felt comfortable, but had just started to feel uncomfortable. The IPD was defined as such in previous studies [3,16,37,39,40]. Subsequently, the distance between the two avatars was transformed at a 1:30 ratio to obtain the psychological IPD [18]. The distance between the two avatars was originally set at 13.33 cm; this indicates an initial distance between the participant and the target of approximately 4 m in the real world [3,16]. The reliability of the measurement used was examined using a pilot study. The intraclass correlation coefficient between the repetitions was 0.85, indicating satisfactory reliability.

### 2.4. Targets

A man and a woman aged 22 years with typical Chinese appearance were selected as targets, with heights of 176.3 and 159.6 cm, respectively. Because the test was conducted in early spring, the targets were dressed in everyday clothing without any accessories. A digital camera (Sony HDR-XR260; Sony, Tokyo, Japan) was used to capture the sagittal view of the man and woman targets under four conditions: two involved mask wearing, and two involved vaccinated status. The targets were requested to maintain a neutral expression during the image capture, and these photographs served as digital targets in the online survey. The heights of the digital male and female targets on the screen were scaled down to 60.0 and 54.3 cm, respectively. The surgical mask used in the test was blue and without any decoration. This was typical of the type of face mask commonly recommended during the COVID-19 pandemic. Unlike the sex and mask variables that could be clearly identified through the images, the text annotations regarding vaccination were superimposed on the bodies of the targets. In the test, participants were instructed that when a target person was labelled as “vaccinated”, indicating that the target had received the complete course of COVID-19 vaccine, as presented in Figure 2.

### 2.5. Procedure

Before data collection, an experimenter explained the testing procedure to the participants. A 2 min video produced by Stanford Medicine was used to introduce the COVID-19 pandemic to the participants to enable them to recall how they felt during the pandemic. Furthermore, the eight images of the targets in 2 × 2 × 2 combinations (Figure 2) were displayed for the corresponding condition before the participants made their perceptual distance judgments. The images were used to help the participants imagine the feeling of facing the target under various situations to ensure IPD data quality. Each participant was requested to complete three trials, and the average values of the trials were calculated for further analysis. A minimum 3 min rest period was provided to each participant between the trials. For IPD judgment, the participants used the mouse to move the avatar to a position where they psychologically felt close to uncomfortable but still comfortable. Each participant had the opportunity to make slight adjustments to the position of their avatar to confirm the perceived distance. Once the participant had determined the IPD, the computer automatically calculated and recorded the distance between the chins of the two avatars [3].

### 2.6. Statistical Analysis

The independent variables in the study were target sex, participant sex, face mask wearing, and vaccination status. The dependent variable was IPD expressed in centimeters. Data were analyzed using SPSS 23.0 (IBM, Armonk, NY, USA), and the significance level (α) was set at 0.05. Four-way analysis of variance (ANOVA) was conducted to evaluate the effect of independent variables on IPD, and the Tukey’s test was used for post hoc comparisons. In addition, two three-way ANOVA were also performed for each participant sex group. The effect size was determined using η^2^ value for each effect. In general, η^2^ = 0.01 indicates a small effect, 0.06 indicates a medium effect, and 0.14 indicates a large effect [41]. Beforehand, the Kolomogorov–Smirnov test was used to verify the compliance of numerical variables with the normal distribution, while Levene’s test was used to verify the homogeneity of variances.

## 3. Results

Through the Kolmogorov–Smirnov test, all of the IPD data collected in the study were found to be normally distributed (D_(800)_ = 0.012, *p* = 0.412), while Levene’s test showed that the data were homogenous (F_(15, 784)_= 1.142, *p* = 0.291). Table 1 presents the four-way ANOVA results for the IPD measurements. All independent variables significantly affected the IPD (all *p* < 0.001), and no interactions were observed among the variables. Figure 3 further illustrates the main effects of the four independent variables. Generally, a relatively short IPD was reported when male participants encountered vaccinated or masked female targets.

Figure 4 presents the varying paired comparisons of the independent variables on IPD between the two vaccinated statuses. As shown in the figure, all pairs exhibited the significant differences (all *p* < 0.05). The greatest IPD was reported when female participants faced unmasked male targets who were not vaccinated (247.5 cm), and the shortest IPD was observed in the opposite situation (112.3 cm). Even though the targets were vaccinated and wearing masks, a relatively long IPD was observed when the female participants encountered male targets (149.8 cm) compared with other sex dyad conditions (approximately 116.5 cm). When averaged across other variables, the effects of mask wearing and vaccination on IPD appeared to be identical (Figure 5), with mean IPD values of 145.7 cm and 148.5 cm, respectively. All paired comparisons also revealed significant differences (all *p* < 0.001).

Before conducting the three-way ANOVA, the normal distribution and homogeneity of variances in IPD data were also confirmed for the subgroups of males (D_(400)_ = 0.011, *p* = 0.183; F_(7, 392)_= 1.483, *p* = 0.149) and females (D_(400)_ = 0.008, *p* = 0.101; F_(7, 392)_= 1.566, *p* = 0.116). Table 2 presents the two three-way ANOVA results for each participant sex group. Target sex seemed not to affect the IPD of male participants (*p* > 0.05), but significantly affected the IPD of female participants (*p* < 0.001).

## 4. Discussion

During the COVID-19 pandemic, frequent hand washing, face mask wearing, and social distancing have become the primary self-protection methods. As the virus continues to spread, vaccine development and the comprehensive administration have also been strongly recommended by public health experts [25,42,43]. However, these preventive measures may affect people’s lifestyle and work. IPD plays a crucial role in human communication and interaction. In this study, online surveys were thus conducted to assess the effects of four factors (i.e., participant sex, target sex, mask wearing, and vaccination) on IPD perception during the COVID-19 pandemic, and the results indicated that all independent variables examined in the study significantly affected the IPD (all *p* < 0.001), and no interaction was observed (Table 1). Compared with investigations on how wearing masks affects the recognized IPD of people, studies on the effect of COVID-19 vaccination on IPD are still lacking. This study was the first to examine the influence of vaccination status on IPD. In the analyses, as hypothesized, the participants may lower the perceived risk level when facing the vaccinated individuals and thus shorting the IPD. Unlike other preventive measures (e.g., mask wearing), vaccination aims to protect individuals from becoming infected and reduce the risk of mortality after diagnosis [28,29,30]. Our findings implied that not only masks, but also vaccination could lead to the shortening of IPD and may cause challenges in the prevention and control of COVID-19 transmission.

As mentioned, the primary concern of this study was the effect of vaccination on IPD. Our findings indicated that the effect of vaccination on IPD reduction (43.5 cm) was only slightly less than that of mask wearing (49.1 cm), as presented in Figure 5. The IPDs reported for wearing and not wearing masks were 145.7 and 194.8 cm, respectively, and those for vaccinated and unvaccinated were 148.5 and 192.0 cm, respectively. When averaged across participant and target sex variables, an unmasked and unvaccinated condition resulted in a greater IPD than the other conditions, with a total difference of 92.7 cm. This may imply that not only mask wearing, but also vaccination could lead to further reductions in IPD; this may pose challenges in the prevention of COVID-19 transmission.

Studies focusing on the influence of mask wearing on IPD during the COVID-19 pandemic have concluded that IPD decreased when participants were faced with a mask-wearing target [1,3,11]. This study obtained a similar result. Cartaud et al. [11] reported that IPD was significantly reduced when targets were wearing a face mask because targets were perceived as being more trustworthy compared with those of the other conditions. Crucially, IPD was further reduced in participants living in low-risk areas, and it was not affected by the predicted health of the targets. Studies have also investigated the effect of sex dyads on IPD; however, the results were inconsistent. Yu et al. [16] observed that male dyads reported the greatest IPDs and female dyads reported the shortest IPDs. These findings were consistent with those of other studies [44,45]. By contrast, Baxter [46] and Evans and Howard [47] revealed that the shortest IPDs were observed in mixed-sex dyads, and Hecht et al. [48] reported that the IPDs reported by mixed-sex dyads were not significantly different from those reported by same-sex dyads. In our study, as presented in Figure 4, when the male participants were facing the female target, a shorter IPD was observed. The average IPDs for male and female dyads in the study were 165.7 and 164.9 cm, respectively (Figure 4). The results of this study were inconsistent with those of Yu et al. [16]. In our study, two mixed-sex dyad combinations were used, namely a male participant with a female target (M-F) and a female participant with a male target (F-M), with reported IPDs of 151.7 and 198.7 cm, respectively. Zhou et al. [49] reported that the social interaction distance between people was related to perceptual judgments concerning social grouping; the female participants tended to maintain a greater distance in mixed-sex dyads because they felt insecure and shy [50]. In our study, the effect of mixed-sex dyads on IPD was unclear because of considerable differences between the F-M and M-F conditions. This can also be explained by the fact that the effect of the target sex on IPD was significant for female participants but nonsignificant for male participants (Table 2).

Although mask wearing and vaccination are still the most effective preventive measures to control the COVID-19 pandemic, these two measures have a negative effect on people’s perception of IPD. In the analyses, mask wearing and vaccination may have caused people to prefer a closer and more risky IPD; this may be consistent with the risk homeostasis theory (RHT) proposed by Wilde [51]. Wilde reported that there was a level of risk that people were generally willing to take. RHT states that, similar to a thermostat with a target temperature, people have a target level of risk and change their behavior to maintain their target level of risk. For instance, driving a car entails a certain level of risk. When this particular behavior is made less risky by the introduction of a seatbelt, the behavior of the driver changes in reaction to the new level of safety. That is, the drivers could perhaps drive faster because they feel that the new level of safety allows for riskier behavior. In this study, when faced with a vaccinated target wearing a mask, participants tended to take a riskier approach (i.e., a shorter IPD) to make the overall risk homeostatic. Si et al. [31] examined 4540 Chinese participants (1825 vaccinated and 2715 unvaccinated), and also reported that vaccination lessened participants’ frequency of hand washing 1.75-fold and their compliance frequency of physical distancing 1.24-fold. However, the rate of mask wearing did not decrease significantly. Their findings indicated that a reduction in the frequency of hand washing and physical distancing could cause a resurgence of COVID-19.

Wearing a mask and maintaining a safe distance do not guarantee self-protection. Studies have reported the numerous risks of contracting COVID-19 even when wearing masks. Sugimura et al. [52] reported that in Hiroshima, Japan, nonmask users were infected at a rate of 16.4% and mask users were infected at a rate of 7.1%, implying that wearing a mask can reduce, but not completely prevent, the risk of contracting COVID-19. Douglas et al. [53] employed a quantitative analysis and suggested that commonly used surgical masks afforded no protection against respirable particles. Sickbert-Bennett et al. [54] also reported that procedural face masks secured with elastic ear loops provided the lowest efficiency, with a fitted filtration efficiency of 38.1%. Moreover, correctly wearing masks is crucial for protective efficiency. Xi et al. [55] indicated that even a small gap of 1 cm^2^ could cause a 17% leakage. A gap area of 4.3  cm^2^ at the nose bridge was the most frequent incorrect practice when wearing a surgical mask, leading to a leakage of 60%. Siahaan et al. [56] conducted a cross-sectional study in Indonesia and reported that only 34.3% of the participants wore face masks correctly. Ganczak et al. [57] observed that the proportion of using masks correctly decreased from 65.9% to 58.9% within 2 weeks. Uncovered noses (47.3–52.7%) and masks around the neck (39.2–42.6%) were the most frequent incorrect practices. Therefore, when the mask is not used correctly and the contact distance is shortened, a false sense of safety may arise, and thus increase the risk of infection. Additionally, because people tend to underestimate the risk when interacting with someone wearing a mask, the importance of using the appropriate masks worn correctly, not reusing masks, and not using cloth or aesthetic masks should be emphasized [1,58].

Masks are a key measure to reduce transmission and save lives. Wearing well-fitted masks should be part of a comprehensive approach including maintaining physical distancing and avoiding crowds and closed and close-contact settings [59]. Masks can be used either for protection of healthy persons or to prevent onward transmission. Even in the presence of patients diagnosed as having COVID-19, masks block the route of infection; however, the function of vaccination in these situations is different. When faced with a vaccinated target, the participants in the study tended to move closer; this may be because the targets were considered relatively safe and healthy. Such a psychological state may be related to the idea that individuals are protected by other people’s vaccinations. This typifies the so-called “Please, you go first!” attitude [33]. In fact, even with vaccination, there is a possibility of COVID-19 reinfection [60]. Reducing the IPD leads to intuitive bias. For the COVID-19 pandemic, the causes and effects of these findings warrant further investigation.

This study has some limitations. A total of 100 young male and female university students were recruited to this study as participants, and caution should be exercised regarding subsequent application of the results obtained from this group to other populations. However, other individual factors affecting the IPD, such as age, education, employment, and the level of exposure to COVID-19 virus, were considered as controlled variables. Because of the pandemic, an online survey was employed. The IPD data obtained may therefore be inconsistent with those obtained in the real world, and face masks may reduce the interpersonal distance in virtual reality [61]. In addition, only blue surgical masks were used in the study. The effects of face masks of various types and colors on IPD perception should be further investigated to provide more information pertaining to social distancing assessment, social interaction enhancement, and environmental design improvement. Moreover, comparisons of results among studies were challenging because the current state of COVID-19 transmission may affect participant perceptions. For instance, compared with the study of Lee and Chen [3], there was an overall increase in IPD; this may be associated with the particular severity of the COVID-19 pandemic during the experimental period in the study. Finally, all participants in this study had received a complete vaccination, and differences in vaccination status may also affect subjective IPD data.

## 5. Conclusions

The effects of participant and target sex, presence or absence of a face mask, and vaccination status on IPD were evaluated in this study. All main effects had a significant influence on IPD determination. In summary, a masked and vaccinated condition led to a shorter IPD than the other conditions, with a total difference of 92.7 cm. This implies that not only masks, but also vaccination could lead to the shortening of IPD. Based on the literature review, in this study, the first investigation was conducted into the influence of vaccination status on IPD. Our findings revealed that the impact of vaccination on IPD was comparable to that of mask usage. However, this result may not benefit the prevention and control of the epidemic, because the primary purpose of vaccination is to provide protection to the recipients rather than solely reducing infection rates. This may cause challenges in the prevention and control of COVID-19 transmission. The results can serve as a reference for government agencies in policy-making and public health promotion.

## Figures and Tables

**Figure 1 healthcare-11-01711-f001:**
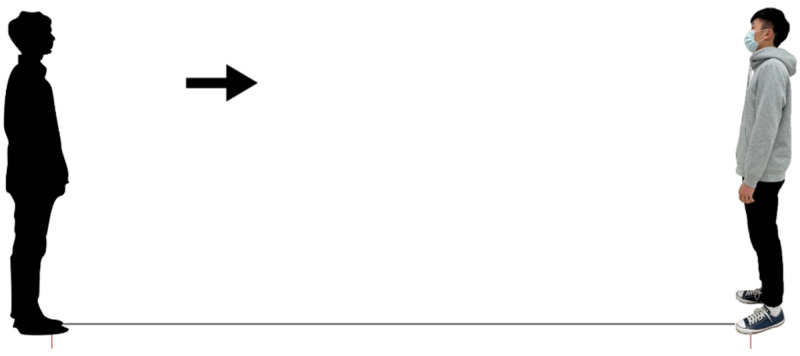
Screenshot of the online survey showing the participant approaching a male target wearing a surgical mask.

**Figure 2 healthcare-11-01711-f002:**
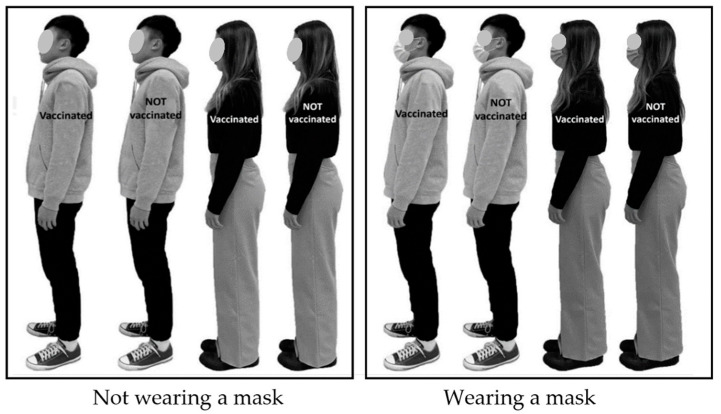
Images of the targets under various testing combinations (2 sexes × 2 mask conditions × 2 vaccinated statuses).

**Figure 3 healthcare-11-01711-f003:**
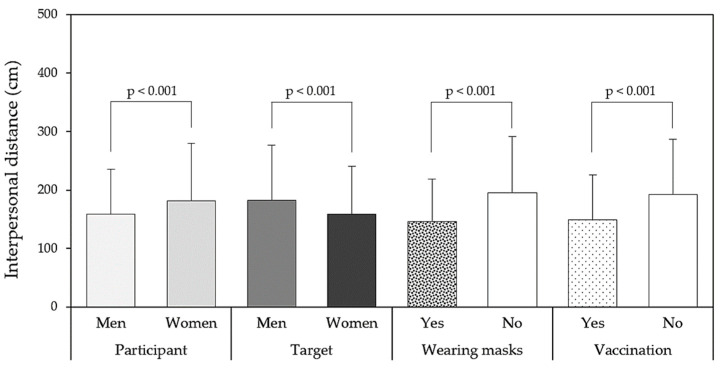
Main effects and the significances of the interpersonal distances for each variable.

**Figure 4 healthcare-11-01711-f004:**
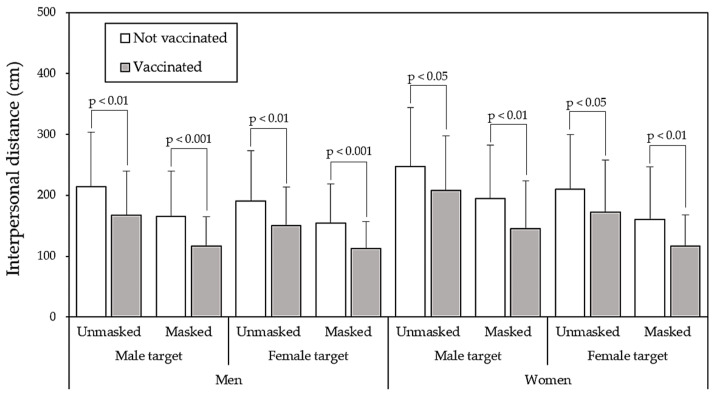
Paired comparisons and the significances of interpersonal distances for each variable between the two vaccination levels.

**Figure 5 healthcare-11-01711-f005:**
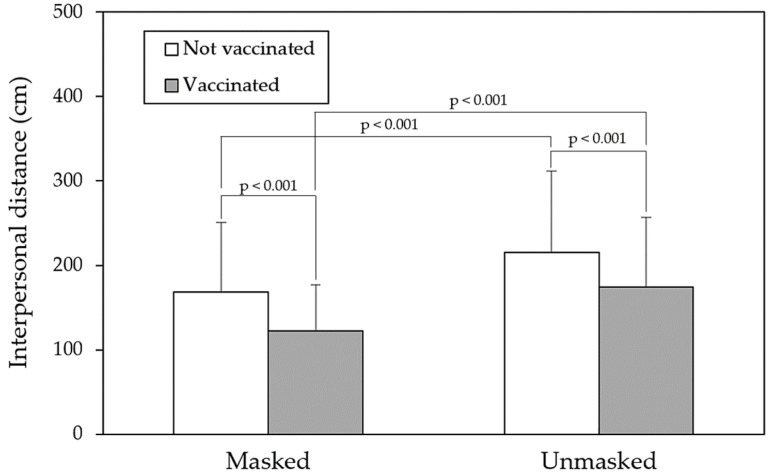
Comparisons of interpersonal distances under various mask wearing and vaccination combinations.

**Table 1 healthcare-11-01711-t001:** Four-way ANOVA results of interpersonal distance.

Sources	SS	df	MS	F	*p*-Value	η^2^
Participant sex (PS)	106,775	1	106,775	15.769	<0.001	0.020
Target sex (TS)	114,183	1	114,183	16.863	<0.001	0.021
Face mask (FM)	482,778	1	482,778	71.300	<0.001	0.083
Vaccinated (V)	379,316	1	379,316	56.020	<0.001	0.067
PS × TS	19,369	1	19,369	2.860	0.091	0.004
PS × FM	7702	1	7702	1.138	0.287	0.001
PS × V	170	1	170	0.025	0.874	<0.001
TS × FM	3972	1	3972	0.587	0.444	0.001
TS × V	1307	1	1307	0.193	0.661	<0.001
WM × V	1313	1	1313	0.194	0.660	<0.001
PS × TS × FM	592	1	592	0.087	0.768	<0.001
PS × TS × V	94	1	94	0.014	0.906	<0.001
PS × FM × V	510	1	510	0.075	0.784	<0.001
TS × FM × V	39	1	39	0.006	0.940	<0.001
PS × TS × FM × V	14	1	14	0.002	0.964	<0.001
Error	5,308,530	784	6771			

**Table 2 healthcare-11-01711-t002:** Three-way ANOVA results on interpersonal distance for each sex group.

Sources	SS	df	MS	F	*p*-Value	η^2^
Men						
Target sex (TS)	19,748	1	19,748	3.844	0.051	0.010
Face mask (FM)	184,260	1	184,260	35.868	<0.001	0.084
Vaccinated (V)	197,783	1	197,783	38.501	<0.001	0.089
TS × FM	3815	1	3815	0.743	0.389	0.002
TS × V	1052	1	1052	0.205	0.651	0.001
FM × V	93	1	93	0.018	0.893	<0.001
TS × FM × V	3	1	3	0.001	0.980	<0.001
Error	2,013,754	392	5137			
Women						
Target sex (TS)	113,803	1	113,803	13.540	<0.001	0.033
Face mask (FM)	306,221	1	306,221	36.433	<0.001	0.085
Vaccinated (V)	181,704	1	181,704	21.618	<0.001	0.052
TS × FM	749	1	749	0.089	0.766	<0.001
TS × V	350	1	350	0.042	0.839	<0.001
FM × V	1730	1	1730	0.206	0.650	0.001
TS × FM × V	49	1	49	0.939	0.939	<0.001
Error	3,294,779	392	8405			

## Data Availability

The data are available upon reasonable request to the Corresponding Author.

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
