# Peer review of "Effects of Target Variables on Interpersonal Distance Perception for Young Taiwanese during the COVID-19 Pandemic"

_healthcare, 2023, doi:10.3390/healthcare11121711_

Round 1
Reviewer 1 Report
Dear authors,
It’s my pleasure to review your work. It’s an interesting topic and study. The attached comments and questions will help your work additionally contribute to the research community.
Kind regards,
Reviewer

Reviewer 2 Report
Overall, the study is very interesting, and the research approach is appropriate. The use of an online survey and avatars makes the study interesting and innovative. The results are interesting and compelling and the comparison of the study's findings with previous research adds knowledge to the area of study (Interpersonal Distance Perception).
My only recommendation for clarity would be to include the research problem, hypothesis, and variables at the beginning of the methods section.
Overall, very good paper, congratulations!
Reviewer 3 Report
The manuscript entitled "Effects of Target Variables on Interpersonal Distance Perception for Young Taiwanese during the COVID-19 Pandemic" is an interesting study and important for further prevention during the pandemic. The introduction is concise but comprehensive. The mothods are described exhaustively, allowing for replication of the study. However, the statistical analysis section should be expanded.
First of all, the sample size was not large, so it is important to check the assumptions for the parametric ANOVA test. If normality and equality of variance is not presented in the sample and subsamples, the alternative non-parametric tests should be used. Please, add these information in details, including how the assumptions were examined (what test was used, e.g., Kolmogorv-Smirnov or Shapiro-WIlk) to the methods section.
Also, the authors stated that they used effect size as a power, which make a big confusion. Please, answer the question how the sample size was a priori determined (and what cut-off was used for a power), for four-way ANOVA (or alternative non-parametric tests). Or, alternatively, what power is demonstrated a posteriori for these tests.
Please report appropriate effect size for ANOVA (e.g., eta-square or partial eta-square) for each effect (or alternative effect size statistic for non-parametric tests).
Table 1 and 2 must include effect size, not power. It is impossible, that df was always 1 for six or eight compared groups. Please check the result and correct all mistakes in Table 1 and 2.
What test was used for a post-hoc comparisons? The authors never mentioned post-hoc tests, so it is completely unclear, how the intergroup differences were found in the study. Please report it in statistical methods subsection as well as in the results.
It would be good to include post-hoc results using p-values in all figures to understand which group is significantly different from the other. Currently, it is completely unclear and figures are not much informative.
The discussion and conclusions should be adequate to new results (especially, if non-parametric tests will be used instead of ANOVA).
Reviewer 4 Report
Dear editors,
Thank you for the opportunity to review the manuscript titled “Effects of Target Variables on Interpersonal Distance Perception for Young Taiwanese during the COVID-19 Pandemic”. In general, the authors have produced a well-structured article that is methodologically adequate and that performs a timely data analysis. In this sense, it could be accepted after making some minor changes that will improve its quality.
Proposed changes:
- State, clearly, the research problem of the study at the end of the introduction.
- Likewise, detail the hypotheses associated with the objective of the study (these must be answered in the conclusions).
- Indicate type of sampling for the selection of participants and inclusion and exclusion criteria.
- Indicate the number or code of the ethics committee that approved it.
- Indicate in the data analysis if the normality of the data was verified (kolmogorov-smirnov).
- After the conclusions, the authors should include a brief reflection on the practical implications obtained in their research work.
Round 2
Reviewer 3 Report
The Authors improved the manuscript, as suggested.